# Changes in Dry Eye Status after Steroid Pulse and Orbital Radiation Therapies in Active Thyroid Eye Disease

**DOI:** 10.3390/jcm11133604

**Published:** 2022-06-22

**Authors:** Yasuhiro Takahashi, Aric Vaidya, Hirohiko Kakizaki

**Affiliations:** 1Department of Oculoplastic, Orbital & Lacrimal Surgery, Aichi Medical University Hospital, Nagakute 480-1195, Japan; aricvaidya1@gmail.com (A.V.); cosme_geka@yahoo.co.jp (H.K.); 2Department of Oculoplastic, Orbital & Lacrimal Surgery, Kirtipur Eye Hospital, Kathmandu 44600, Nepal

**Keywords:** active phase, thyroid eye disease, steroid pulse therapy, orbital radiation therapy, dry eye, meibomian gland dysfunction

## Abstract

This prospective, observational study examined changes in dry eye status after steroid pulse and orbital radiation therapies in 16 patients (32 eyes) with active thyroid eye disease (TED). TED status was evaluated through clinical activity score (CAS), margin reflex distance (MRD)-1 and 2, presence or absence of Graefe’s sign/lid lag, and Hertel exophthalmometric value. Dry eye status was quantified through presence or absence of superior limbic keratoconjunctivitis, corneal fluorescein staining (AD score), tear break-up time, Schirmer test I results, tear meniscus height, and dry eye-related quality of life score. Meibomian gland dysfunction (MGD) was evaluated through Marx line score, eyelid abnormalities (MGD score), meibum expression score, and meibomian gland loss score. Those items were measured before and 6 months after treatment, and the results were statistically compared. Consequently, CAS significantly improved, and MRD-1 significantly decreased after treatment (*p* < 0.050). Although a part of MGD status improved (*p* < 0.050), all items regarding dry eye status did not change significantly after treatment (*p* > 0.050). Steroid pulse and orbital radiation therapies did not largely alter most items regarding dry eye and MGD status.

## 1. Introduction

Dry eye frequently develops in patients with thyroid eye disease (TED), with a prevalence of 65–85% [1]. Several mechanisms for the development of dry eye are proposed in TED, including reduced aqueous tear production due to lacrimal gland involvement, excess tear evaporation due to increased ocular surface exposure, meibomian gland dysfunction (MGD) due to incomplete blinking, the inflammatory process of TED inducing ocular surface damage, and abnormal friction between the ocular surface and eyelid due to increased eyelid pressure [1,2,3,4,5,6,7,8,9]. Eyelid retraction and proptosis cause increased ocular surface exposure and eyelid pressure, and incomplete blinking [4,6,7,8]. Patients in the active phase of TED show more severe manifestations of dry eye compared to those in the inactive phase [1,4].

Steroid pulse and orbital radiation therapies are anti-inflammatory treatment modalities for active TED [10]. These can reduce periocular inflammation and improve eyelid retraction and proptosis, which may possibly provide indirect improvement of dry eye status. Contrarily, orbital radiation therapy occasionally causes new-onset dry eye or deteriorates dry eye status [11,12]. However, none of the previous studies had compared the dry eye status before and after steroid pulse and orbital radiation therapies.

Here, we examined the dry eye status before and after steroid pulse and orbital radiation therapies and compared the measurement results.

## 2. Materials and Methods

### 2.1. Study Design and Patients

This was a prospective, observational study including Japanese patients with TED who underwent both steroid pulse and orbital radiation therapies from October 2014 to December 2015. A diagnosis of TED was made based on the presence of at least one characteristic sign (eyelid fullness, eyelid retraction, proptosis, and/or restrictive strabismus) and the presence of thyroid autoimmunity [7]. All patients were in the active phase of TED, confirmed by high-intensity lesions in at least one extraocular muscle shown on fat-suppressed coronal T2-weighted magnetic resonance images (Figure 1). None of the patients had undergone previous eyelid or orbital surgery, orbital radiotherapy, or had any history of contact lens use. All patients showed patent lacrimal drainage system confirmed by lacrimal syringing.

### 2.2. Measurements

All measurements were performed by one oculoplasty specialist (Y.T.) before and 6 months after steroid pulse and orbital radiation therapies.

#### 2.2.1. TED Condition

Clinical activity score (CAS), margin reflex distance (MRD)-1 and 2, presence or absence of Graefe’s sign/lid lag, and Hertel exophthalmometric value were assessed. CAS was calculated using 7 parameters: retrobulbar discomfort, pain on eye movement, eyelid erythema, eyelid swelling, conjunctival injection, chemosis, and caruncle swelling [13]. MRD-1 and MRD-2 were measured as the distance from the upper (MRD-1) or lower (MRD-2) eyelid margin to the corneal light reflex in the primary eye position. The patient was set in the sitting position with brow fixation and was requested to look at a light source (a penlight), then the distances were recorded using a millimeter ruler. Presence or absence of Graefe’s sign and lid lag was confirmed in each eyelid. Graefe’s sign is a dynamic phenomenon wherein the affected eyelid lags behind on downward rotation of the eye [14]. Lid lag is a static condition in which the affected eyelid is higher than the normal while the eye is in downgaze [14]. In Hertel exophthalmometry measurements, the distance from the corneal apex to a plane defined by the deepest point on the lateral orbital rim was measured. A base value, the distance between the 2 footplates, was documented at the first measurement and reproduced at the follow-up examinations.

#### 2.2.2. General Assessment of Dry Eye

Presence or absence of superior limbic keratoconjunctivitis (SLK), the area (A) and density (D) classification of corneal fluorescein staining, tear break-up time (TBUT), Schirmer test I, tear meniscus height (TMH), and dry eye-related quality of life score (DEQS) were used to assess the severity of dry eye. Patients with at least 2 of the following criteria were diagnosed with SLK: blood vessel dilation in the superior bulbar conjunctiva, papillary inflammation of the upper tarsal conjunctiva, punctate fluorescein staining of the superior conjunctiva and the upper cornea, filaments in the upper cornea, epithelial thickening of the superior bulbar conjunctiva, and redundancy of the superior bulbar conjunctiva [8]. The AD classification was graded using the scale reported by Miyata et al. (Table 1) [15]. The TBUT was determined by fluorescein staining of the ocular surface. The time just after eye-opening to the first appearance of a dry spot on the cornea was measured. Schirmer test I was performed without anesthesia as follows: a Schirmer test strip was placed in the lower conjunctival sac without touching the cornea, and the length of the wet portion after 5 min was measured. TMH was measured on the sagittal plane through the center of the upper and lower eyelids using optical coherence tomography (RS-3000, NIDEK Co., Ltd., Aichi, Japan) [16]. The dedicated attachment was used to observe the anterior segment of the eyes. All patients filled in the 15-item DEQS questionnaire [17]. It consists of 2 subscales, the “bothersome effects of ocular symptoms (6 items)” and the “impact of dry eye on daily life (9 items)” scores. The DEQS asks patients for the frequency (A-column) and severity of each item (B-column). We calculated the DEQS score as follows: the sum of scores in the B-column × number of valid responses × 25. All the scores ranged from 0 to 100, with a higher score representing greater disability.

#### 2.2.3. Assessment of MGD

MGD was assessed through 4 criteria: the position of Marx line; the presence or absence of eyelid abnormalities; the quality and ease of meibum expression; the loss of the meibomian glands.

The position of the Marx line was determined by fluorescein staining of the ocular surface. The resulting stained lines along the eyelids were examined by slit-lamp biomicroscopy after several blinks. The grading scale reported by Yamaguchi et al. was used (Table 1) [18]. The eyelid was further divided into 3 segments (the outer third, middle third, and inner third) and the position of the Marx line was evaluated in each; the maximum score for each eyelid was 9. The presence or absence of eyelid abnormalities (irregular lid margin, vascular engorgement, plugged meibomian orifices, and foaming of tear meniscus) was expressed using a binary system (a dummy variable; 0 = absence, 1 = presence), and total sum scores (MGD score) were calculated in each upper and lower eyelid for statistical analysis. The maximum MGD score was 4 points. The grading of quality and ease of meibum expression that we previously defined was used (Table 1) [16]. The loss of meibomian glands in the eyelids was evaluated using a mobile pen-shaped meibography (Meibom Pen; Japan Focus Co., Ltd.; Tokyo, Japan). The grading scale previously reported by Arita et al. was used (Table 1) [19].

### 2.3. Treatment

#### 2.3.1. Steroid Pulse Therapy

All patients underwent 3 cycles of steroid pulse therapy under regimen of body weight × 10 mg/kg/day of methylprednisolone for 3 days per cycle under hospitalization. Patients did not receive subsequent oral prednisolone [10].

#### 2.3.2. Orbital Radiation Therapy

Patients were immobilized in the supine position using a thermoplastic mask, and a computed tomography scan with a slice thickness of 5 mm or less was taken for planning treatment [10]. Both the target area and orbital structures for which irradiation was avoided were contoured using treatment planning systems (XiO; Electa, CMS, St Louis, MO, USA or Eclipse; Varian Medical System, Palo Alto, CA, USA) [10]. Clinical target volume was defined utilizing the extraocular muscles and retrobulbar soft tissue, and the planning target volume was defined as clinical target volume with an additional 0.5 cm margin [10]. Irradiation was delivered to the planning target volume using 6-MV photons, with lateral opposing fields angled posteriorly at 2–4° to align the anterior field edges [10]. Although a 0.5 cm multi-leaf collimator margin was utilized, the anterior–posterior margin was reduced to spare the lens and pituitary gland [10]. A total dose of 20 Gy in 10 fractions was delivered to all patients, during hospitalization for steroid pulse therapy [10].

### 2.4. Statistical Analyses

Patient age and measurement results are expressed as the mean value ± standard deviation. The measurement results were compared before and after treatment using paired t-test or Wilcoxon signed-rank test. The ratios of presence or absence of Graefe’s sign, lid lag, and SLK before and after treatment were compared using Fisher’s exact test. All statistical analyses were performed using SPSS™ ver. 26 software (IBM Japan, Tokyo, Japan). A *p*-value of <0.050 was considered statistically significant.

## 3. Results

Patient demographic data are shown in Table 2. This study included 32 eyes from 16 patients (3 males and 13 females; mean age, 60.2 ± 10.5 years). Three patients were excluded from this study because of loss of follow-up. Three patients had undergone steroid pulse therapy at another clinic previously, and 2 patients had diabetes mellitus. Three patients were current smokers at the time of steroid pulse and orbital radiation therapies (10 cigarettes/day in one patient; 15 cigarettes/day in one patient; and 20 cigarettes/day in the other patient). Bilateral dysthyroid optic neuropathy developed in 4 patients.

The measurement results are shown in Table 3. Regarding TED status, CAS significantly improved after treatment (*p* = 0.001). MRD-1 significantly decreased after treatment (*p* = 0.046). The other pre-treatment measurement results did not change after treatment (*p* > 0.050). In regard to dry eye status, all items did not change after treatment (*p* > 0.050). Concerning MGD status, we could not examine meibomian gland loss in the upper eyelid in 3 patients because of the difficulty of upper eyelid eversion (Gifford’s sign). The Marx line score in the upper eyelid (*p* = 0.060), meibum expression score in the upper eyelid (*p* = 0.034), and meibomian gland loss score in the upper eyelid (*p* = 0.025) slightly improved after treatment. The other measurement items did not change after treatment (*p* > 0.050).

## 4. Discussion

We compared the dry eye status before and after steroid pulse and orbital radiation therapies. Although a part of MGD status improved, the degrees of the changes were small. In addition, most items regarding dry eye and MGD status did not change after treatment. This indicates that anti-inflammatory treatment for active TED does not alter the dry eye status.

The autoimmune inflammatory process of TED is thought to cause ocular surface damage [9]. Previous studies showed more severe dry eye and MGD status in patients with active TED, compared to those with inactive TED [3,4,9,20,21,22]. However, although the CAS decreased after treatment, dry eye status did not change remarkably. Prolonged ocular surface inflammation may cause irreversible ocular surface damage and lacrimal gland involvement, which may be reflected in the results of this study.

Upper eyelid retraction caused by an enlarged levator palpebrae superioris muscle and proptosis produce taut upper eyelid pressure, which induce abnormal friction between ocular surface and palpebral conjunctiva, and ocular surface inflammation [5,6,7,8]. This leads to deterioration of dry eye, SLK, and MGD [5,6,7,8]. A previous study showed improvement of SLK and upper eyelid retraction, and reduction in upper eyelid pressure in patients with TED who underwent orbital decompression [8]. In the present study, although anti-inflammatory treatment lowered upper eyelid position, dry eye condition and the ratio of patients with SLK did not change largely after treatment. A small change in upper eyelid position and no significant improvement of proptosis after treatment may be causes of this discrepancy.

Marx line, Meibum expression, and meibomian gland loss scores slightly improved after steroid pulse and orbital radiation therapies. Meibomian glands secrete lipids at normal or enhanced level to compensate lipid secretion from residual meibomian glands [23]. Thereafter, obstructive MGD gradually progresses and finally, irreversible atrophic changes and glandular dropout occur in meibomian glands [23,24]. In this study, pre-treatment Marx line, MGD, and meibomian gland loss scores were not so high, indicating a slight chance for mild restoration of meibomian gland function.

Orbital radiation therapy itself can cause new-onset dry eye or deteriorates dry eye status [11,12,25,26]. Proposed mechanisms of this complication are depletion of goblet cells and damage on the lacrimal and meibomian glands [25]. The incidence of dry eye ranges from 47 to 81% after orbital radiotherapy for tumors [25,26], while that accounts for 4–12% after orbital irradiation for TED [11,12]. This study also demonstrated little changes in dry eye status after orbital radiation therapy. One of the possible reasons was a small dose of radiation therapy, compared to treatment for tumors. This may enable secretary cells/glands to tolerate toxicity of radiation therapy. A short follow-up period may also be a cause of slight changes in dry eye status in this study. Moreover, deterioration of dry eye status after orbital radiation may offset improvement of dry eye status due to subsidence of periocular inflammation.

All examinations were performed by a single examiner. Although this may cause an examiner bias in this study, a part of the measurements is subjective. We believe that the measurements by a single examiner was valid for this study.

Our study was limited by several factors. Inclusion of a small number of patients was the main flaw of this study. Additionally, we lost follow-up in 3 patients. Variability in smoking habit is another limitation of this study. We did not evaluate the morphological changes in the lacrimal and meibomian glands and thickness of tear film lipid layer, which may provide more information.

## 5. Conclusions

In conclusion, we performed statistical comparison of dry eye status before and after steroid pulse and orbital radiation therapies. Although a part of MGD status improved, steroid pulse and orbital radiation therapies did not largely alter most items regarding dry eye and MGD status.

## Figures and Tables

**Figure 1 jcm-11-03604-f001:**
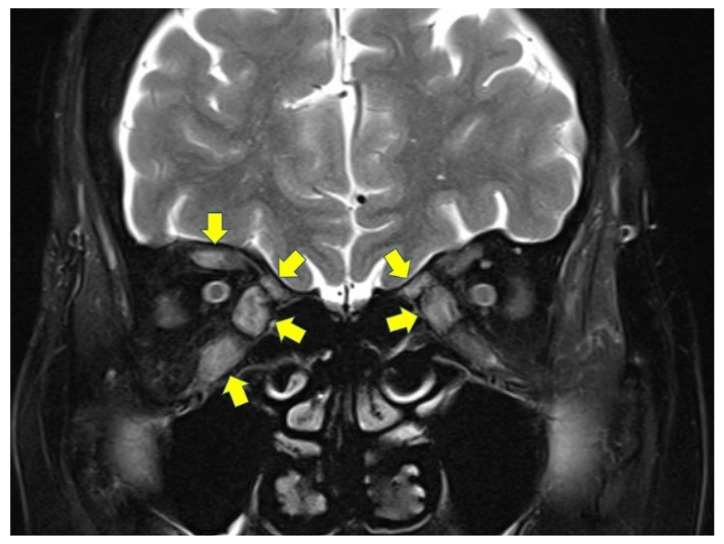
A fat-suppressed coronal T2-weighted magnetic resonance image indicating inflammation in multiple extraocular muscles (arrows).

**Table 1 jcm-11-03604-t001:** The grading systems of measurement parameters.

Measurement Parameters	Grading	Findings
A	0	No punctate staining
	1	The staining involving less than one-third of the cornea
	2	The staining involving one-third to two-thirds of the cornea
	3	The staining involving more than two-thirds of the cornea
D	0	No punctate staining
	1	Sparse density
	2	Moderate density
	3	High density and overlapped lesions
Marx line	0	The line runs entirely along the conjunctival side of the meibomian gland orifices
	1	Parts of the line touch the meibomian orifices
	2	The line runs through the meibomian orifices
	3	The line runs along the eyelid margin side of the meibomian orifices
Meibum expression	0	Easy expression of clear meibum with mild eyelid compression
	1	Cloudy meibum expression with mild compression
	2	Cloudy meibum expression with moderate compression
	3	Toothpaste-like meibum expression with more than moderate compression
	4	No expression even with hard compression
Meibography score	0	No loss of meibomian glands
	1	Area loss less than one-third of the total meibomian gland area
	2	Area loss between one-third and two-thirds of total meibomian gland area
	3	Area loss more than two-thirds of total meibomian gland area

**Table 2 jcm-11-03604-t002:** The demographic data.

Patient number	16
Sex (M/F)	3/13
Age (years)	60.2 ± 10.5
Period from development of ocular symptoms to treatment (months)	8.9 ± 10.7
Medical history	
Steroid treatment	3
DM	2
Smokers	3
DON	4

M, male; F, female; DM, diabetes mellitus; DON, dysthyroid optic neuropathy.

**Table 3 jcm-11-03604-t003:** Comparison of results measured before and after treatment.

	Pre-Treatment	Post-Treatment	*p* Value
CAS	3.3 ± 2.0	0.1 ± 0.3	0.001 *
Median (IQR)	3.5 (1.25–4.75)	0 (0–0)	
MRD-1 (mm)	4.4 ±1.6	4.2 ± 1.6	0.046 ^†^
MRD-2 (mm)	5.8 ± 1.0	5.8 ± 0.9	1.000 ^†^
Graefe’s sign (Y/N)	19/13	15/17	0.226 ^‡^
Lid lag (Y/N)	5/27	3/29	0.354 ^‡^
Hertel exophthalmometry (mm)	17.4 ± 2.8	16.9 ± 3.3	0.158 ^†^
SLK (Y/N)	6/26	10/22	0.194 ^‡^
AD classification			
A	0.7 ± 0.7	1.0 ± 0.9	0.162 *
Median (IQR)	0 (0–1.0)	0 (0–1.0)	
D	1.5 ± 1.4	1.5 ± 1.3	0.829 *
Median (IQR)	1.5 (0–3.0)	1.0 (0–3.0)	
Schirmer test (mm)	14.1 ± 10.6	13.8 ± 12.0	0.878 ^†^
TBUT (s)	1.5 ± 0.8	1.9 ± 1.7	0.157 ^†^
TMH (μm)			
Upper eyelid	295.9 ± 68.8	301.6 ± 216.9	0.889 ^†^
Lower eyelid	364.1 ± 224.8	349.6 ± 219.0	0.787 ^†^
DEQS	41.0 ± 21.5	37.6 ± 29.6	0.636 ^†^
Marx line score			
Upper eyelid	5.8 ± 1.8	5.0 ± 2.3	0.060 *
Median (IQR)	6.0 (6.0–7.0)	6.0 (3.0–7.0)	
Lower eyelid	5.9 ± 1.8	6.1 ± 1.4	0.598 *
Median (IQR)	6.0 (6.0–7.0)	6.5 (6.0–7.0)	
MGD score			
Upper eyelid	1.6 ± 1.1	1.8 ± 1.2	0.438 *
Median (IQR)	2.0 (1.0–3.0)	2.0 (1.0–3.0)	
Lower eyelid	1.6 ± 1.1	1.9 ± 1.1	0.231 *
Median (IQR)	2.0 (1.0–2.0)	2.0 (1.0–3.0)	
Meibum expression score			
Upper eyelid	2.7 ± 1.5	2.1 ± 1.6	0.034 *
Median (IQR)	3.0 (1.0–4.0)	1.5 (1.0–4.0)	
Lower eyelid	2.5 ± 1.5	2.1 ± 1.5	0.134 *
Median (IQR)	3.0 (1.0–4.0)	2.0 (1.0–4.0)	
Meibomian gland loss score			
Upper eyelid (13 patients)	1.6 ± 0.8	1.4 ± 0.8	0.025 *
Median (IQR)	1.0 (1.0–2.0)	1.0 (1.0–2.0)	
Lower eyelid	0.6 ± 0.9	0.6 ± 0.8	0.739 *
Median (IQR)	0 (0–1.0)	0 (0–1.0)	

CAS, clinical activity score; IQR, interquartile range; MRD, margin reflex distance; Y, yes; N, no; SLK, superior limbic keratoconjunctivitis; TBUT, tear break-up time; TMH, tear meniscus height; DEQS, Dry Eye-Related Quality-of-Life Score; MGD, meibomian gland dysfunction. * Wilcoxon signed-rank test; ^†^ paired *t*-test; ^‡^ Fisher’s exact test.

## Data Availability

Not applicable.

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
