# Peer review of "Changes in Dry Eye Status after Steroid Pulse and Orbital Radiation Therapies in Active Thyroid Eye Disease"

_jcm, 2022, doi:10.3390/jcm11133604_

Round 1

Reviewer 1 Report

Dear Authors:
The study is interesting but has an important limitation which is. 
 the sample size that you assume at the end of the manuscript. It is true that the prevalence of this type of conditions is not high but the small sample size, the loss of some patients and the variability in habits such as smoking make the results inconsistent.  
Another important limitation also referred to in the manuscript is that clinical observations are often subjective depending on the observer, it should be ensured that the same physician is the evaluator of these clinical signs. 

Author Response

Dear Authors:

The study is interesting but has an important limitation which is the sample size that you assume at the end of the manuscript. It is true that the prevalence of this type of conditions is not high but the small sample size, the loss of some patients and the variability in habits such as smoking make the results inconsistent. 

Reply: Thank you very much for your comment, and we agree with your comment. We had stated the inclusion of small number of patients as a limitation in the original version, and we newly added that loss of follow-up in 3 patients and variability in smoking habit are limitations of this study (the last paragraph of the “Discussion” section). 

Another important limitation also referred to in the manuscript is that clinical observations are often subjective depending on the observer, it should be ensured that the same physician is the evaluator of these clinical signs.

Reply: Thank you very much for your comment. All measurements were performed by one oculoplasty specialist. We stated it in page 2, the “Measurements” sub-section. In addition, we added the following sentences in the 6th paragraph of the “Discussion” section: All examinations were performed by a single examiner. Although this may cause an examiner bias in this study, a part of the measurements is subjective. We believe that the measurements by a single examiner was valid for this study.

Reviewer 2 Report

Takahashi et al. describe a prospective observation of 16 subjects (32 eyes). They analyzed changes in TED and DE status after steroid and radiation therapies for thyroid disease. The paper is incredibly well-written, topical, and interesting. In addition, the methods are sound and support their results and conclusions.

I have the following minor suggestions for readability, etc.:

1. Line 10: I believe "32 eyes" is more standard than "32 sides"

2. Line 34: This is VERY minor, but I suggest removing the comma here -- "...of dry eye, compared to those..."

3. Lines 60-61: COMMENT ONLY -- Some readers/other reviewers may object to the use of only one observer. I am fine with it, but--while acknowledged in Lines 210-212--the authors may want to justify only using one observer.

4. Line 70: English v. American English -- suggest "penlight" (vs. "pen torch")

5. Table 2: Please clear up this table. Is "Past history" the heading for for "Steroid Tx, DM, etc."?

6. Line 160: MAJOR (please change) -- "(P < 0.050)" should be "(P > 0.050)"

7. Line 164: "On the contrary..." should read "Conversely..."

8. Table 3: Two comments:

   A) If possible, can the authors describe which variables were normal (i.e., used t-tests) and which were not (i.e., used Mann-Whitney)? In the table, the non-normal ones PROBABLY should have IQRs instead of S.D.s as measures of spread.

  B) Are these correct?: Please check --

     Marx line score
Upper eyelid  4.7 ± 2.3  5.4 ± 2.0  0.060 <--is this correct?
Lower eyelid  4.7 ± 2.3  5.4 ± 2.0  0.598  <--is this correct?

Meibum expression score
Upper eyelid 3.1 ± 1.2 3.0 ± 1.3 0.034
Lower eyelid 3.0 ± 1.3 3.0 ± 1.3 0.134

Thank you for allowing me to reviewer this interesting and topical manuscript.

Author Response

Takahashi et al. describe a prospective observation of 16 subjects (32 eyes). They analyzed changes in TED and DE status after steroid and radiation therapies for thyroid disease. The paper is incredibly well-written, topical, and interesting. In addition, the methods are sound and support their results and conclusions.

I have the following minor suggestions for readability, etc.:

  1. Line 10: I believe "32 eyes" is more standard than "32 sides"

Reply: Thank you very much for your suggestion. We changed the term according to your suggestion. In addition, we also changed “sides” to “eyes” in the second sentence of the “Results” section.

  1. Line 34: This is VERY minor, but I suggest removing the comma here -- "...of dry eye, compared to those..."

Reply: Thank you very much for your suggestion. We removed the comma.

  1. Lines 60-61: COMMENT ONLY -- Some readers/other reviewers may object to the use of only one observer. I am fine with it, but--while acknowledged in Lines 210-212--the authors may want to justify only using one observer.

Reply: Thank you very much for your suggestion. As commented by reviewer #1, a part of the measurements is subjective. The measurements by a single examiner may be, therefore, valid for this study. We added these contents in the 6th paragraph of the “Discussion” section.

  1. Line 70: English v. American English -- suggest "penlight" (vs. "pen torch")

Reply: Thank you very much for your suggestion. We changed the term “pen torch” to “penlight”.

  1. Table 2: Please clear up this table. Is "Past history" the heading for "Steroid Tx, DM, etc."?

Reply: Yes. We used indents for steroid treatment and DM for understandability of the heading “medical history” for these items.

  1. Line 160: MAJOR (please change) -- "(P < 0.050)" should be "(P > 0.050)"

Reply: We apologize for the mistake. We corrected it.

  1. Line 164: "On the contrary..." should read "Conversely..."

Reply: Thank you very much for your suggestion. According to the comment #8 from you, we corrected the results and deleted the term “on the contrary”.

  1. Table 3: Two comments:
  2. A) If possible, can the authors describe which variables were normal (i.e., used t-tests) and which were not (i.e., used Mann-Whitney)? In the table, the non-normal ones PROBABLY should have IQRs instead of S.D.s as measures of spread.
  3. B) Are these correct?: Please check --

     Marx line score

Upper eyelid  4.7 ± 2.3  5.4 ± 2.0  0.060 <--is this correct?

Lower eyelid  4.7 ± 2.3  5.4 ± 2.0  0.598  <--is this correct?

Meibum expression score

Upper eyelid 3.1 ± 1.2 3.0 ± 1.3 0.034

Lower eyelid 3.0 ± 1.3 3.0 ± 1.3 0.134

Thank you for allowing me to reviewer this interesting and topical manuscript.

Reply: Thank you very much for your indication. We compared ordinal variables, such as CAS, AD score, Marx line score, meibum expression score, MGD score, and meibomian gland loss score, using Wilcoxon signed-rank test. We added medians and IQRs in Table 3.

We apologize for the errors. A part of numerical values was recognized as character strings in the EXCEL file, which was the cause of error in calculating mean values and SD. We corrected all the errors in Table 3. 

Reviewer 3 Report

I consider this manuscript to have valuable data that would be of interest. However, I have some doubts that needs to beclarify. 1. What kind of surgery was performed to patients ? Page 5 , line 165. There is only information about steroid puls therapy and radiation therapy in the materials and methods section... 2. The radiation therapy itself is the cause of dry eye syndrom so I suppose that we have two oppsite mechanism of action: improving of TED (by both therapies) is decreasing should decrease dry eye symptoms while radaiation therapy  increases the dry eye. The authors should discuss this or propose other explanation why the dry eye syndrom is not decreased when TED improves ? 3. The conculsions that anti-inflammatory therapy does not affect dry eye syndrom is not supported by the study results i.e. you mau only conclude that combination of steroid pulses together with radiation therapy  did not affect your study group.

Author Response

I consider this manuscript to have valuable data that would be of interest. However, I have some doubts that needs to be clarify.

  1. What kind of surgery was performed to patients? Page 5, line 165. There is only information about steroid pulse therapy and radiation therapy in the materials and methods section...

Reply: We apologize for the mistake. We did not perform any surgery during the study period. We changed the term “surgery” to “treatment”.

  1. The radiation therapy itself is the cause of dry eye syndrome so I suppose that we have two opposite mechanism of action: improving of TED (by both therapies) is decreasing should decrease dry eye symptoms while radiation therapy increases the dry eye. The authors should discuss this or propose other explanation why the dry eye syndrome is not decreased when TED improves?

Reply: Thank you very much for your comment, and we agree with your comment. We think that deterioration of dry eye status after orbital radiation may offset improvement of dry eye status due to subsidence of periocular inflammation. We added this content in the 5th paragraph of the “Discussion” section, the last sentence.

  1. The conclusions that anti-inflammatory therapy does not affect dry eye syndrome is not supported by the study results i.e. you may only conclude that combination of steroid pulses together with radiation therapy did not affect your study group.

Reply: Thank you very much for your comment. We changed the corresponding sentence as follows: Although a part of MGD status improved, steroid pulse and orbital radiation therapies did not largely alter most items regarding dry eye and MGD status.

Round 2

Reviewer 1 Report

Dear authors

The sample size is  still the main limitation to support the results but it´s true that it´s difficult to recruit patients with this orbitopathy .

Author Response

Thank you very much for your comment. We added the following sentence “Inclusion of a small number of patients was the main flaw of this study.” in the 7th paragraph of the Discussion section.

Reviewer 3 Report

However, I am satisfied with the changes made by authors, the conclusions in the abstract needs re-writing as was done in main text.

Author Response

Thank you very much for your suggestion. We changed the conclusion of the abstract, based on the change in the main text.